# Therapeutic Plasma Exchange for the Treatment of Hyperthyroidism: Approach to the Patient with Thyrotoxicosis or Antithyroid-Drugs Induced Agranulocytosis

**DOI:** 10.3390/jpm13030517

**Published:** 2023-03-13

**Authors:** Irene Tizianel, Chiara Sabbadin, Simona Censi, Cristina Clausi, Anna Colpo, Anca Irina Leahu, Maurizio Iacobone, Caterina Mian, Carla Scaroni, Filippo Ceccato

**Affiliations:** 1Department of Medicine (DIMED), University of Padova, Via Giustiniani 2, 35128 Padova, Italy; 2Endocrinology Unit, Padova University Hospital, Via Ospedale Civile 105, 35128 Padova, Italy; 3Department of Transfusion Medicine, Padua University Hospital, Via Giustiniani 2, 35128 Padova, Italy; 4Endocrine Surgery Unit, Department of Surgery, Oncology and Gastroenterology (DISCOG), Padua University Hospital, Via Giustiniani 2, 35128 Padova, Italy

**Keywords:** therapeutic plasma exchange, hyperthyroidism, thyrotoxicosis, antithyroid-drugs, agranulocytosis

## Abstract

Primary hyperthyroidism is an endocrine disorder characterized by excessive thyroid hormone synthesis and secretion by the thyroid gland. Clinical manifestations of hyperthyroidism can vary from subclinical to overt forms. In rare cases, hyperthyroidism may represent a clinical emergency, requiring admission to an intensive care unit due to an acute and severe exacerbation of thyrotoxicosis, known as a thyroid storm. First-line treatment of hyperthyroidism is almost always based on medical therapy (with thioamides, beta-adrenergic blocking agents, glucocorticoids), radioactive iodine or total thyroidectomy, tailored to the patient’s diagnosis. In cases of failure/intolerance/adverse events or contraindication to these therapies, as well as in life-threatening situations, including a thyroid storm, it is necessary to consider an alternative treatment with extracorporeal systems, such as therapeutic plasma exchange (TPE). This approach can promptly resolve severe conditions by removing circulating thyroid hormones. Here we described two different applications of TPE in clinical practice: the first case is an example of thyrotoxicosis due to amiodarone treatment, while the second one is an example of a severe adverse event to antithyroid drugs (agranulocytosis induced by methimazole).

## 1. Introduction

Excessive thyroid hormone synthesis and secretion by the thyroid gland characterizes hyperthyroidism, a common endocrine disorder. Clinical manifestations can vary from subclinical to overt forms [1]. In rare cases, hyperthyroidism may develop as a clinical emergency, requiring hospitalization or intensive care unit (ICU) admission, especially if an acute and severe exacerbation of thyrotoxicosis leads to a thyroid storm. The prevalence of hyperthyroidism in Western Countries is 1–3%; a thyroid storm occurs in 0.22% of all thyrotoxic patients [2]: it is a rare but life-threatening condition, with a high mortality rate (10–30%) due to multiorgan failure (MOF). It is characterized by hyperpyrexia, severe tachycardia/atrial fibrillation, heart failure, neurological and gastrointestinal-hepatic dysfunction. The diagnostic process is difficult due to the absence of defined clinical and biochemical criteria. It is essential to recognize this condition early to ensure prompt therapy and provide specific management, if necessary [3]. In cases of underlying conditions or frailty conditions, such as heart failure or atrial fibrillation, a thyroid storm can precipitate the clinical course to MOF.

The first line of treatment for hyperthyroidism is almost always based on medical therapy with thioamides (methimazole and propylthiouracil), beta-adrenergic blocking agents, glucocorticoids, radioactive iodine and total thyroidectomy (Figure 1) [4]. In cases of autoimmune hyperthyroidism, current medical treatments are not effective in the pathogenesis of Graves’ disease. Although some immunomodulating properties of thioamides have been reported, most antithyroid drugs exert their effects through the inhibition of thyroid peroxidase, the enzyme involved in thyroid hormone synthesis [5]. Methimazole is the classical and most widely distributed thionamide; propylthiouracil is the least potent compound (10-fold weaker than methimazole), but it additionally inhibits the T4 to T3 in peripheral tissues [6].

In case of failure/intolerance or contraindication of these therapies, as well as in life-threatening situations (including thyroid storm or therapies-induced severe adverse events), therapeutic plasma exchange (TPE) has been proposed as an alternative treatment, based upon extracorporeal systems [7]. This approach can promptly resolve severe conditions by removing circulating thyroid hormones [8]. TPE was first used in the treatment of hyperthyroidism in the 1970s by Ashkar et al. on three cases of thyroid storm, and its role is still a matter of debate [9]. It is mainly performed with albumin as replacement fluid, and it can be a bridge therapy before definitive surgery. It works by reducing protein-bound thyroid hormones, autoantibodies and cytokines and removing 5′ monodeiodinase, converting T4 to T3 from the circulation. However, these effects are transient and last for 24–48 h; therefore, a rebound of thyrotoxicosis is expected after these 1–2 days [4].

In this review, we described two different applications of TPE in clinical practice: the first case is thyrotoxicosis due to amiodarone treatment, while the second one is an example of a severe adverse event of antithyroid drugs, i.e., agranulocytosis induced by methimazole.

## 2. Clinical Case Presentation

### 2.1. Case 1: Amiodarone-Induced Hyperthyroidism

In October 2021, a 52-year-old patient went to the emergency room for palpitations and dyspnea; his medical history included atrial fibrillation, previously treated with two electrical cardioversions and amiodarone in 2017, arterial hypertension, and obesity (BMI 35 kg/m^2^). He was treated with continuous infusion of amiodarone for rate control; on biochemical exams, his TSH value was suppressed. The patient was then referred for an Endocrine consultation. At the ultrasound, the thyroid appeared to be diffusely hypoechoic, with a mild heterogeneous echotexture, with an absence of vascularization and nodules (Figure 2); his heart rate (HR) was arrhythmic at 100 beats per minute (bpm). Laboratory analysis revealed a suppressed value of TSH and significantly increased levels of thyroid hormones: FT4 100 pmol/L (reference range 9–22) and FT3 20.9 pmol/L (3.6–6.8); anti-TSH receptor antibodies (TRAB) were negative. Therefore, based on these clinical features, he was diagnosed with amiodarone-induced thyrotoxicosis type 2 (AIT-2) versus a mixed form, and he was then hospitalized at our Endocrine Unit.

The initial treatment was: methimazole 30 mg daily, prednisone 25 mg daily and propranolol 40 mg 4 times daily. Amiodarone was discontinued, and digoxin (0.25 mg daily) was considered to normalize HR.

He was discharged three days later, and after twenty days, he undertook a cardiological examination, which showed persistent atrial fibrillation with a high ventricular rate; therefore, he was hospitalized again at our Unit. During admission, he appeared anxious with hyperhidrosis and upper limbs tremor; he had no fever, HR was arrhythmic at 140–150 bpm, and blood pressure values were 150/90 mmHg. Laboratory analysis still revealed thyrotoxicosis (FT4 100 pmol/L, FT3 41 pmol/L, suppressed TSH). The electrocardiogram confirmed atrial fibrillation: mean HR 135 bpm, normal corrected QT interval (QTc) at 420 mSec; the trans-thoracic echocardiography excluded heart failure with preserved ejection fraction (72%) and without left atrial enlargement.

Antithyroid therapy was switched to propylthiouracil 200 mg twice daily, being more effective in blocking the T4 to T3 conversion, and dexamethasone 4 mg twice daily. He was also treated with sodium perchlorate (10 oral drops twice daily) and propranolol (80 mg twice daily). According to the American Thyroid Association (ATA) guidelines for the management of hyperthyroidism and thyrotoxicosis, thyrotoxic patients who are undergoing thyroidectomy should be rendered euthyroid (with medical treatment) before undergoing surgery because a thyroid storm may be precipitated by the stress of surgery, anesthesia, or thyroid manipulation [10]. Moreover, a longer preoperative treatment reduces intraoperative bleeding, allowing better thyroid visualization and preservation of the nerves and parathyroid glands [11].

Amiodarone, a type III antiarrhythmic lipophilic drug, has high iodine content (75 mg of iodine per 200 mg tablet) [12], and it is associated with various side effects. AIT can be distinguished into type-1 (AIT-1) and type-2 (AIT-2): AIT-1 results from an increased synthesis of T4 and T3 due to the iodine overload in a preexisting thyroid disease. On the contrary, AIT-2 results from an excessive release of preformed thyroid hormone due to destructive thyroiditis, with no thyroid alteration before amiodarone therapy; mixed forms are caused by both these mechanisms [13]. During amiodarone treatment, it is recommended to assess thyroid function every 3–6 months to ensure early detection of some relevant biochemical alterations [14]. Although AIT is less frequent than amiodarone-induced hypothyroidism (AIH), this condition is more severe than AIH since it can evolve into a thyroid storm. It is mandatory to promptly recognize the subtype of AIT in order to provide the most effective treatment [15].

### 2.2. Case 2: Methimazole-Induced Agranulocytosis

In June 2019, a 46-year-old patient went to the emergency room for palpitations, weight loss of about 5 kg in the last 2–3 weeks and asthenia. He had no personal or familial history of thyroid disease; he did not take any medication and had no known allergies. The biochemical exams showed suppressed TSH (0.00 miU/L), increased FT4 and FT3 (100 pmol/L and 43 pmol/L, respectively), positive TRAB (11 UI/L) and negative anti-TPO. At the ultrasound, the thyroid appeared diffusely enlarged; echostructural and vascular patterns were suggestive of Graves’ disease (as depicted in Figure 3).

Following diagnosis, high-dose methimazole (30 mg daily) and propranolol (20 mg twice daily) were started. The biochemical exams performed after one month showed severe leukopenia and increased liver function tests. A few days later, he complained of a fever (40 °C) and sore throat, so he went back to the emergency room. His body temperature was 39.9 °C, blood pressure values were 130/80 mmHg, and heart rate was 70 bpm. His physical examination did not reveal any alteration, such as thorax x-ray, abdominal ultrasound and video laryngoscopy. The exams showed low white blood cell count with severe neutropenia (0.85 × 10^9^/L), increased levels of C-reactive protein (150 mg/L) and procalcitonin (2.66 ug/L). A diagnosis of methimazole-induced agranulocytosis was made, and he was admitted to our Endocrine Unit.

The therapeutic management included discontinuation of methimazole, daily administration of granulocyte-colony stimulating factors (G-CSF) to normalize blood count formula, and a broad-spectrum of antimicrobic therapy with piperacillin/tazobactam, vancomycin and caspofungin. Despite the antibiotic therapy, the patient had a persistent fever that was thought to be secondary to hyperthyroidism and not to neutropenia; after normalization of the white blood count, steroids were added to block the conversion of FT4 to FT3.

Agranulocytosis is defined as an absolute neutrophil count < 0.5 × 10^9^/L and is one of the most serious adverse effects of antithyroid drugs. Antithyroid drugs are usually well tolerated, especially at the beginning of the treatment course; large doses can cause some minor side effects, such as pruritus, itching, and skin reactions [5]. The incidence is about 0.1–0.5%, and it is thought to be an immune-mediated process resulting in peripheral leukocyte destruction [16]. Agranulocytosis is usually observed early (within one to three months after initiation of therapy), can be dose-related, and is more frequent in older patients. It may even occur during a second or third course of the same treatment [17,18]. An immunofluorescence test with anti-neutrophil antibodies revealed transient granulocyte-specific IgG, induced by propylthiouracil [19], with complement-dependent lysis that affected both mature blood cells and bone precursor cells [20]. Antineutrophil cytoplasmic antibodies may contribute to agranulocytosis [21].

Agranulocytosis may also cause severe infections, which should be treated by broad-spectrum antibiotics and G-CSF [22]. Patients must be warned to pay attention to sore throat, fever or both and immediately perform a blood count, discontinue medications and refer to the physician. Agranulocytosis usually occurs within 90 days from the beginning of therapy, and it is indicated to obtain a complete recovery of blood count before starting treatment [1]. Thionamides (propylthiouracil and methimazole) have a known immunological cross-reactivity that can induce agranulocytosis; therefore, if this adverse effect appears with one of these drugs, the other one is contraindicated [23].

## 3. Clinical Presentation and Management of Severe Hyperthyroidism

The diagnosis of a thyroid storm relies on clinical manifestations, not endocrine assessment. A thyroid storm is the presentation of severe thyrotoxicosis: the activation of (high) thyroid hormones and (normal) catecholamines act synergistically. A high clinical index of suspicion for thyroid storm should be maintained in all patients with thyrotoxicosis (most cases occur following a precipitating event or intercurrent illness), especially when associated with any initial evidence of systemic decompensation [10], in order to prevent multiorgan failure. Effective management relies on prompt recognition of early clinical manifestations because the high concentration of thyroid hormones is not a diagnostic criterion for a thyroid storm.

Some risk factors may contribute to complicating severe hyperthyroidism and precipitating a thyroid storm; moreover, the excess of free thyroid hormones predisposes the patient and speeds up the clinical picture. Thyroid surgery, a common precipitant in the past, has become a relatively rare trigger because most patients are promptly treated with antithyroid drugs before thyroidectomy. Other known events not associated with thyroid illness or irregular use of antithyroid drugs are infections, diabetic ketoacidosis, pulmonary thromboembolism, severe emotional stress, or trauma [24,25]. In clinical practice, a patient with a precarious balance of clinical conditions is at high risk. Therefore, a careful clinical follow-up is suggested to prevent a thyroid storm. In the authors’ opinion, since endocrine levels are not able to predict the development of a thyroid storm, an effort to increase the awareness of patients to recognize early signs/symptoms of decompensation should be conducted, as in the early management of an Addisonian crisis [26]. Moreover, in order to reduce the gap between patients and Healthcare Providers, a dedicated e-mail address and a facility for teleconsultation may reduce the risk that unconsidered thyrotoxicosis can evolve into a thyroid storm [27].

During a thyroid storm, an exaggerated response to catecholamines dominates the manifestations of thyrotoxicosis: thyroid hormones increase the cellular response to catecholamines, and catecholamines increase peripheral conversion of T4 to T3 [28,29]. Common presenting symptoms of thyroid storm are tachycardia (up to atrial fibrillation), congestive heart failure, elevated temperature, hyperdynamic circulation, central nervous system effects (anxiety, tremors, insomnia) and gastrointestinal dysfunction [30]. Despite intensive care management, a thyroid storm is still characterized by an increased mortality rate (up to 20%) [31]. There are some clinical tools that combine several parameters to predict the need for aggressive therapy: the most used are the Burch–Wartofsky Point Scale and the Japanese Thyroid Association scale [31]. Both scores combine several clinical parameters such as thermoregulatory dysfunction; tachycardia or atrial fibrillation; signs/symptoms of congestive heart failure; gastrointestinal-hepatic dysfunction or central nervous system manifestations; and an ongoing precipitating event [32].

The goals of severe thyrotoxicosis/thyroid storm management are to reverse systemic decompensation with hospitalization (as soon as possible, in ICU if required), to block the peripheral action of thyroid hormones and inhibit thyroid hormone secretion. To achieve a quick normalization of the thyroid function, first-line medical treatment includes antithyroid drugs that inhibit thyroid peroxidase (propylthiouracil and methimazole); short-term preoperative iodine solutions (via the “Wolff-Chaikoff effect”) after thionamides; potassium perchlorate in case of high iodine load (as amiodarone); glucocorticoids as hydrocortisone or dexamethasone; and high-dose propranolol [29,31]. Acetaminophen (paracetamol) is the preferred treatment for thyrotoxicosis-related fever [31].

In cases of severe refractory thyrotoxicosis unresponsive to medical therapy, a plasmapheresis is a useful option for normalizing thyroid function. TPE has been used in several different diseases, all associated with a “high” concentration of a molecule (antibodies, immune complexes, toxins, drugs, hormones) that needs to be lowered. Plasma exchange is a therapeutic process in which a colloid (the most used are albumin and/or plasma) or a combination of crystalloid/colloid replacement solution is infused back, instead of patient plasma, after the plasma extraction from other components of the blood [33]. In physiological conditions, free thyroid hormones bind to thyroxine-binding globulin (TBG), transthyretin and albumin. The proportion of albumin-bound hormones rises during hyperthyroidism: the TBG-bound thyroid hormones are removed with the plasma during TPE. Then, the use of plasma as a replacement fluid provides new free binding sites for free thyroid hormones. Moreover, TPE also removes deiodinase, resulting in reduced conversion of T4 to T3 [34]. Therefore, thyroid hormones should be measured daily during TPE. Other indications of TPE in hyperthyroidism, not explained in this manuscript, are severe symptoms and rapid clinical deterioration; contraindications to medical therapy; liver disease; molar pregnancy; and Graves’ disease-related severe ophthalmopathy [35].

Adverse events or complications during or after TPE are described in 27% of procedures in large series, especially hypotension in predisposed anemic patients (8% of the whole cohort) and minor allergic reactions related to the replacement fluids (chills and fever, skin rash, nausea) [36]. TPE-induced coagulopathy, hypocalcemia, vascular injury, tetany, and seizure are rare reactions. Potentially life-threatening complications occurred during 1% of TPE: since a thyroid storm can be a challenging clinical situation, the scales tip in favor of the procedure in cases of severe thyrotoxicosis.

## 4. TPE in Hyperthyroidism

### 4.1. Case One

Despite maximal medical therapy, the patient was still symptomatic, especially for tachyarrhythmia, with persistent thyrotoxicosis (Table 1). Total thyroidectomy was planned after scheduling TPE sessions to obtain clinical stabilization and an initial reduction of thyroid hormone concentration. After the placement of a double-lumen central venous catheter, the patient underwent three TPE sessions with progressive improvement of thyroid hormonal levels. The replacement fluid was albumin in TPE sessions one and two; in the third TPE session, performed the day before surgery, 30% of fresh frozen plasma (FFP) was also used in order to reduce the risk of hemorrhage due to the depletion of coagulation factors caused by TPE. He was also considered at a high prothrombotic risk due to obesity, immobility after positioning of a central venous catheter in the femoral vein, atrial fibrillation and hyperthyroidism; given the need for TPE, he was switched from oral anticoagulant therapy to low molecular weight heparin, according to his body weight.

He then underwent a total thyroidectomy without any complications and preservation of his parathyroid glands. Four days after surgery, he developed hypothyroidism (low fT3, fT4 still above normal range) and levothyroxine treatment was started. During the follow-up, thyroid function was adequately compensated by substitutive therapy with thyroxine, and atrial fibrillation was well controlled without digoxin therapy.

### 4.2. Case Two

Four days after discontinuation of methimazole, the patient presented a severe recurrence of hyperthyroidism (Table 2). Since the patient was still neutropenic and pyretic, the use of propylthiouracil was contraindicated, and a total thyroidectomy was not recommended. For these reasons, the indication for TPE to treat hyperthyroidism was discussed with hematologists and Apheresis Specialists. Overall, the patient performed four TPE sessions, and after thyroid hormonal normalization, he was no more pyretic, and thyrotoxicosis symptoms gradually improved. At this point, the patient underwent a total thyroidectomy with an uneventful intraoperative and postoperative course. As in the previous case, FFP was used in the last TPE session performed before surgery. Three days after surgery, he developed hypothyroidism; thus, levothyroxine therapy was initiated.

## 5. Discussion

TPE can be used for the treatment of hyperthyroidism when standard treatments are ineffective, not tolerated or contraindicated, such as in these two cases [37,38]. It represents an alternative option before surgery, as a bridge therapy to the definitive treatment.

TPE is indicated in category II of treatment (“Disorders for which apheresis is accepted as second-line therapy, either as a standalone treatment or in conjunction with other modes of treatment”) in patients with thyroid storm/hyperthyroidism, as reported in the 2019 guidelines of the American Society for Apheresis [33]. The rationale for TPE lies in the removal of T4 and T3 proteins, reduction of T4 conversion, and the clearance of amiodarone in case of AIT: given amiodarone’s long half-life (60–142 days [15]), TPE is used to lower amiodarone plasma concentrations in patients with AIT, particularly in those without thyroid disease that develop destructive thyroiditis. A recent systematic review of the literature by Garla et al. described the different applications of TPE in thyroid diseases. TPE was used in sixteen cases not responding to the conventional therapy, thirteen cases of agranulocytosis or other thioamides-induced side effects, and eight cases of AIT [37]. TPE also has a role in the clearance of cytokines, deiodinase enzymes, and Graves’ auto-antibodies, with possible improvements also in Graves’ ophthalmopathy and pretibial myxedema [4].

The major contraindications to Therapeutic Apheresis treatments are hemodynamic instability, sepsis, and a history of allergy to the substitution fluid (albumin or fresh frozen plasma). As previously stated, TPE is generally a safe treatment, with only 0.36% of serious adverse events in an Italian registry study [39], especially if performed by experienced staff. Hypocalcemic reactions are one of the most frequent complications of TPE: these reactions can be effectively prevented by a continuous infusion of calcium gluconate during treatment. 

During TPE, an important volume of plasma is exchanged (usually 1-1.5 of Total Plasma Volume), with the purpose of removing antibodies and thyroid hormones bound to proteins from the circulation. Usually, more than one session of TPE is required to control severe hyperthyroidism: a reduction of 50% of free thyroid hormones is an expected result [38]. TPE is usually well tolerated; severe adverse events have been reported only in 0.6% of treatments [39]. When albumin is used as a replacement fluid, the removal of coagulation factors occurs, mainly fibrinogen, and this must be considered in cases of surgery. In our cases, fresh frozen plasma was used as part of the replacement fluid in the last session before surgery.

Case one is an example of the failure of medical therapy in an AIT2 versus mixed form. Severe thyrotoxicosis, unresponsive to standard treatment, can evolve into a thyroid storm, requiring an alternative and effective treatment in order to perform surgery in safe conditions. The patient had no personal history of thyroid disease, but he was at a serious risk of severe heart failure due to atrial fibrillation and high ventricular frequency, in spite of the multiple pharmacological treatments failing to promptly restore a euthyroid status. In the literature, few similar cases described the application of TPE in patients with AIT, cardiac failure, or dangerous arrhythmias [3,40,41].

Case two is an example of agranulocytosis induced by high-dose methimazole therapy. Given the immunological cross-reactivity, the use of propylthiouracil was contraindicated. Moreover, the persistence of a neutropenic fever prevented the possibility of performing a total thyroidectomy for the associated anesthesiologic and surgical risk. It was mandatory to control hyperthyroidism before surgery in safe clinical conditions, and TPE was an effective solution. Agranulocytosis is an absolute indication of urgent hospitalization and immediate discontinuation of thioamides, and the mortality rate can reach 4%. In most cases, recovery is obtained with steroids, G-CSFs and broad-spectrum antibiotics; the onset of severe infections up to sepsis or septic shock worsen clinical conditions and make it difficult to use standard treatments [42,43].

## 6. Conclusions

When medical therapy is ineffective, the final treatment of severe hyperthyroidism is radioactive iodine or surgery. However, in particular conditions of severe hyperthyroidism with clinical complications, it can be necessary to perform TPE before total thyroidectomy in order to perform safe surgery during stable clinical conditions [38]. In conclusion, these cases provide useful examples of TPE application in different thyroid diseases. TPE is an effective alternative treatment before surgery in patients with severe hyperthyroidism in which standard therapy is not effective or contraindicated [44,45].

## Figures and Tables

**Figure 1 jpm-13-00517-f001:**
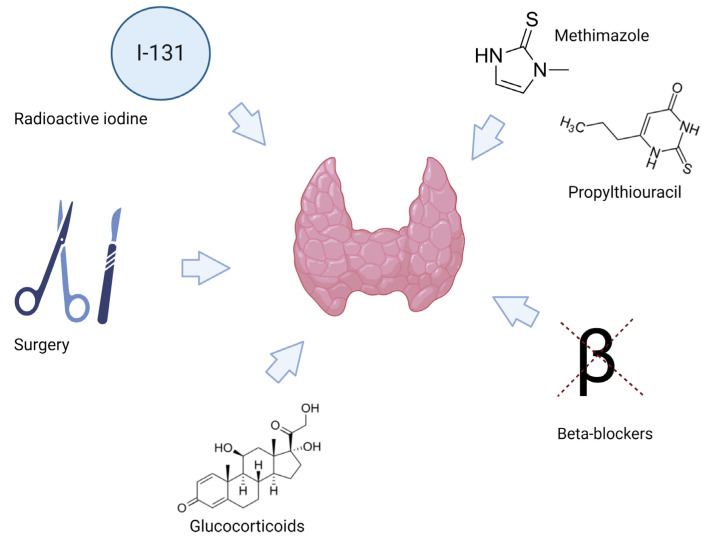
Treatment options for hyperthyroidism. Created with biorender.com (accessed on 3 February 2023).

**Figure 2 jpm-13-00517-f002:**
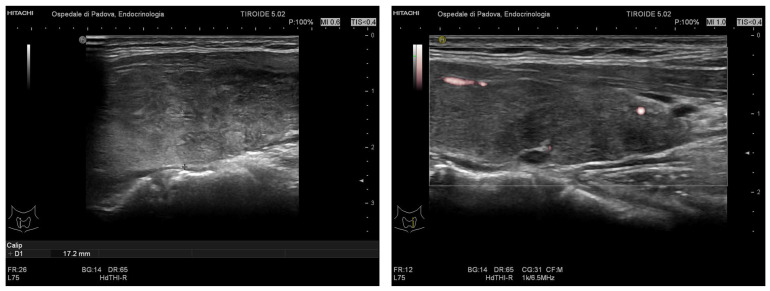
Thyroid ultrasound of Case 1 (right and left lobe, the latter with vascularization).

**Figure 3 jpm-13-00517-f003:**
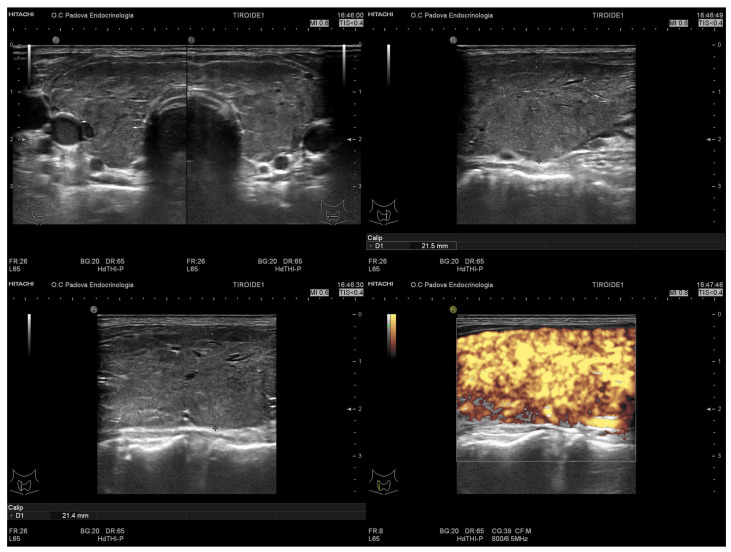
Thyroid ultrasound of Case 2.

**Table 1 jpm-13-00517-t001:** Thyroid hormone status from the diagnosis in patient 1.

Day from Diagnosis	TSH (0.2–4 mIU/L)	FT4 (9–22 pmol/L)	FT3 (3.6–6.8 pmol/L)	Treatment
1	0.01	100	20.9	Methimazole 30 mg/die, prednisone 25 mg/die propranolol 40 mg × 4/die, and stop amiodarone
20	0.01	100	41	Propylthiouracil 200 mg × 2/die, dexamethasone 4 mg × 2/die; sodium perchlorate 10 gtt × 2/die, digoxin 0.25 mg/die, propranolol 80 mg × 2/die
22	-	100	34.9	Before 1st TPE
25	0.01	100	6.93	After 2nd TPE sessions
27	0.01	100	3.98	After 3rd TPE sessions → total thyroidectomy
28	-	61.9	2.29	On discharge from our department
32	0.01	23.3	1.89	Start of levothyroxine treatment
158	0.9	16.23	4.66	Last follow-up visit

TSH: thyroid stimulating hormone; FT4: free tetraiodothyronine; FT3: free triiodothyronine; TPE: therapeutic plasma exchange.

**Table 2 jpm-13-00517-t002:** Thyroid hormone status from the diagnosis in patient 2.

Day from Diagnosis	TSH (0.2–4 mIU/L)	FT4 (9–22 pmol/L)	FT3 (3.6–6.8 pmol/L)	Neutrophils (4.4–11 × 10^9^/L)	White Blood Count (1.8–7.8 × 10^9^/L)	Treatment
1	0.01	100	43.95	-	4.32	Start of methimazole 30 mg/die with progressive decalage, and propranolol 20 mg × 2/die
33	0.01	23.42	7.46	0.01	0.85	Hospital admission
34	-	-	-	0.00	0.89	Granulocyte-colony stimulation factors, broad-spectrum antibiotic therapy
37	-	85.27	35.75	1.43	3.06	Granulocyte-colony stimulation factors, broad-spectrum antibiotic therapy
39	-	-	-	3.37	5.15	Add of prednisone 50 mg/die
41	-	28.87	4.62	-	-	After 4 TPE sessions
45	-	17.9	3.98	-	-	After total thyroidectomy
46	0.69	15.64	2.57	-	-	At discharge

TSH: thyroid stimulating hormone; FT4: free tetraiodothyronine; FT3: free triiodothyronine; TPE: therapeutic plasma exchange.

## Data Availability

Not applicable.

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
