# Peer review of "Therapeutic Plasma Exchange for the Treatment of Hyperthyroidism: Approach to the Patient with Thyrotoxicosis or Antithyroid-Drugs Induced Agranulocytosis"

_jpm, 2023, doi:10.3390/jpm13030517_

Round 1
Reviewer 1 Report
Overall, it is a good manuscript, although I have some comments:
Line 18, 44: “almost based” could be expressed as “almost always based on..”
Line 57: “this 1-2 days” should be replaced by “these 1-2 days”
Case 1: More details are needed. The diagnosis of thyroid storm in the first case is not properly supported by the presented clinical features, as the described signs and symptoms support only the diagnosis of a severe thyrotoxicosis (normal temperature, no data regarding cardiac evaluation- clinical, ultrasound, no gastrointestinal manifestations, no severe CNS involvement, no other laboratory data- liver function tests, bilirubin, glycemia, etc). As the diagnosis of thyroid storm is based on very severe symptoms (cardiovascular life-threatening manifestations as heart failure, pulmonary edema, hyperpyrexia, altered mentation, etc), the diagnosis should be made very carefully. The concentration of thyroid hormones are not diagnostic criteria for thyroid storm, especially in patients with Amiodarone induced-thyrotoxicosis, as the drug’s half-life is very long and the thyrotoxicosis can even worsen in the first month after interruption of the drug.
Even in severe amiodarone- induced thyrotoxicosis, if the patient present severe tachyarrhythmia, TPE is useful for rapidly decreasing the thyroid hormones concentration.
Author Response
REVIEWER NUMBER 1
Overall, it is a good manuscript, although I have some comments:
Line 18, 44: “almost based” could be expressed as “almost always based on..”
[reply]: thanks for the suggestion.
Line 57: “this 1-2 days” should be replaced by “these 1-2 days”
[reply]: we apologize for the error, we have fixed it.
Case 1: More details are needed. The diagnosis of thyroid storm in the first case is not properly supported by the presented clinical features, as the described signs and symptoms support only the diagnosis of a severe thyrotoxicosis (normal temperature, no data regarding cardiac evaluation- clinical, ultrasound, no gastrointestinal manifestations, no severe CNS involvement, no other laboratory data- liver function tests, bilirubin, glycemia, etc). As the diagnosis of thyroid storm is based on very severe symptoms (cardiovascular life-threatening manifestations as heart failure, pulmonary edema, hyperpyrexia, altered mentation, etc), the diagnosis should be made very carefully. The concentration of thyroid hormones are not diagnostic criteria for thyroid storm, especially in patients with Amiodarone induced-thyrotoxicosis, as the drug’s half-life is very long and the thyrotoxicosis can even worsen in the first month after interruption of the drug.
Even in severe amiodarone- induced thyrotoxicosis, if the patient present severe tachyarrhythmia, TPE is useful for rapidly decreasing the thyroid hormones concentration.
[reply]: we agree with your valuable suggestion: our patient presents with increased thyroid hormones, with elevated hearth rate, nonetheless there were not signs or symptoms of cardio-pulmonary failure. Moreover, we added more data regarding cardiological manifestation.

Reviewer 2 Report
Dear Authors,
Thanks for the opportunity to review your paper
The presentation of clinical cases is exhaustive and ,also if the topics is rare, is a perfect example of collaboration between endocrinologist, anestesiologist and surgeons.
I have some questions to ask you.
- Are there some risk factor that may contribute to these complications ?
-I thinks that these patients were under clinical follow-up. Is it possible to avoid these complications with a more strict clinical control?
- What can be the contraindications and related complications to use the Therapeutic plasma exchange?
I think that is useful to integrate this informations in your paper
Author Response
REVIEWER NUMBER 2
Dear Authors,
Thanks for the opportunity to review your paper
The presentation of clinical cases is exhaustive and ,also if the topics is rare, is a perfect example of collaboration between endocrinologist, anestesiologist and surgeons.
I have some questions to ask you.
- Are there some risk factor that may contribute to these complications ?
- I thinks that these patients were under clinical follow-up. Is it possible to avoid these complications with a more strict clinical control?
- What can be the contraindications and related complications to use the Therapeutic plasma exchange?
I think that is useful to integrate this informations in your paper
[reply]: thanks so much for the points raised, we added the required data in order to increase the awareness of the clinical condition that can be related to thyroid storm and its management. We added risk factors that can precipitate a crisis, some suggestions regarding a possible prevention, and the complications related to TPE.
